

# Spectral Bridges

## Scalable Spectral Clustering free from hyperparameters

ISSN 2824-7795

Félix Laplante    Université de Paris Saclay

Christophe Ambroise [1]    Laboratoire de Mathématiques et Modélisation d'Evry, Université Paris-Saclay, CNRS, Univ Evry,

Date published: 2025-07-06    Last modified: 2024-07-09

**Abstract**

In this paper, Spectral Bridges, a novel clustering algorithm, is introduced. This algorithm builds upon the traditional k-means and spectral clustering frameworks by subdividing data into small Voronoï regions, which are subsequently merged according to a connectivity measure. Drawing inspiration from Support Vector Machine's margin concept, a non-parametric clustering approach is proposed, building an affinity margin between each pair of Voronoï regions. This approach is characterized by minimal hyperparameters and delineation of intricate, non-convex cluster structures.

The numerical experiments underscore Spectral Bridges as a fast, robust, and versatile tool for sophisticated clustering tasks spanning diverse domains. Its efficacy extends to large-scale scenarios encompassing both real-world and synthetic datasets.

The Spectral Bridge algorithm is implemented both in Python (https://pypi.org/project/spectral-bridges) and R https://github.com/cambroise/spectral-bridges-Rpackage).

*Keywords:* spectral clustering, vector quantization, scalable, non-parametric

# Contents

---

[1]Corresponding author: christophe.ambroise@univ-evry.fr

# 1 Introduction

Clustering is a fundamental technique for exploratory data analysis, organizing a set of objects into distinct homogeneous groups known as clusters. It is extensively utilized across various fields, such as biology for gene expression analysis (Eisen et al. 1998), social sciences for community detection in social networks (Latouche, Birmelé, and Ambroise 2011), and psychology for identifying behavioral patterns. Clustering is often employed alongside supervised learning as a pre-processing step, helping to structure and simplify data, thus enhancing the performance and interpretability of subsequent predictive models (Verhaak et al. 2010). Additionally, clustering can be integrated into supervised learning algorithms, such as mixture of experts (Jacobs et al. 1991), as part of a multi-objective strategy.

There are various approaches to clustering, and the quality of the results is largely determined by how the similarity between objects is defined, either through a similarity measure or a distance metric. Clustering techniques originate from diverse fields of research, such as genetics, psychometry, statistics, and computer science. Some methods are entirely heuristic, while others aim to optimize specific criteria and can be related to statistical models.

Density-based methods identify regions within the data with a high concentration of points, corresponding to the modes of the joint density. A notable non-parametric example of this approach is DBSCAN (Ester et al. 1996). In contrast, model-based clustering, such as Gaussian mixture models, represents a parametric approach to density-based methods. Model-based clustering assumes that the data is generated from a mixture of underlying probability distributions, typically Gaussian distributions. Each cluster is viewed as a component of this mixture model, and the Expectation-Maximization (EM) algorithm is often used to estimate the parameters. This approach provides a probabilistic framework for clustering, allowing for the incorporation of prior knowledge and the

ability to handle more complex cluster shapes and distributions (McLachlan and Peel 2000).

Geometric approaches, such as k-means (MacQueen et al. 1967), are distance-based methods that aim to partition data by optimizing a criterion reflecting group homogeneity. The k-means++ algorithm (Arthur and Vassilvitskii 2006) enhances this approach by providing faster and more reliable results. However, a key limitation of these methods is the assumption of linear boundaries between clusters, implying that clusters are convex. To address non-convex clusters, the kernel trick can be applied, allowing for a more flexible k-means algorithm. This approach is comparable to spectral clustering in handling complex cluster boundaries (Dhillon, Guan, and Kulis 2004). The k-means algorithm can also be interpreted within the framework of model-based clustering under specific assumptions (Govaert and Nadif 2003), revealing that it is essentially a special case of the more general Gaussian mixture models, where clusters are assumed to be spherical Gaussian distributions with equal variance.

Graph-based methods represent data as a graph, with vertices symbolizing data points and edges weighted to indicate the affinity between these points. Spectral clustering can be seen as a relaxed version of the graph cut algorithm (Shi and Malik 2000). However, traditional spectral clustering faces significant limitations due to its high time and space complexity, greatly hindering its applicability to large-scale problems (Von Luxburg 2007).

The method we propose aims to find non-convex clusters in large datasets, without relying on a parametric model, by using spectral clustering based on an affinity that characterizes the local density of the data. The algorithm described in this paper draws from numerous clustering approaches. The initial intuition is to detect high-density areas. To this end, vector quantization is used to divide the space into a Voronoï tessellation. An original geometric criterion is then employed to detect pairs of Voronoï regions that are either distant from each other or separated by a low-density boundary. Finally, this affinity measure is considered as the weight of an edge in a complete graph connecting the centroids of the tessellation, and a spectral clustering algorithm is used to find a partition of this graph. The only parameters of the algorithm are the number of Voronoï Cells and the number of clusters.

The paper begins with a section dedicated to presenting the context and related algorithms, followed by a detailed description of the proposed algorithm. Experiments and comparisons with reference algorithms are then conducted on both real and synthetic data.

## 2 Related Work

Spectral clustering is a graph-based approach that computes the eigen-vectors of the graph's Laplacian matrix. This technique transforms the data into a lower-dimensional space, making the clusters more discernible. A standard algorithm like k-means is then applied to these transformed features to identify the clusters (Von Luxburg 2007). Spectral clustering enables capturing complex data structures and discerning clusters based on the connectivity of data points in a transformed space, effectively treating it as a relaxed graph cut problem.

Classical spectral clustering involves two phases: construction of the affinity matrix and eigen-decomposition. Constructing the affinity matrix requires $O(n^2 d)$ time and $O(n^2)$ memory, while eigen-decomposition demands $O(n^3)$ time and $O(n^2)$ memory, where $n$ is the data size and $d$ is the dimension. As $n$ increases, the computational load escalates significantly (Von Luxburg 2007).

To mitigate this computational burden, one common approach is to sparsify the affinity matrix and use sparse eigen-solvers, reducing memory costs but still requiring computation of all original matrix entries (Von Luxburg 2007). Another strategy is sub-matrix construction. The Nyström method randomly selects $m$ representatives from the dataset to form an $n \times m$ affinity sub-matrix (Chen et al. 2010). Cai et al. extended this with the landmark-based spectral clustering method, which uses

k-means to determine $m$ cluster centers as representatives (Cai and Chen 2014). Ultra-scalable spectral clustering (U-SPEC) employs a hybrid representative selection strategy and a fast approximation method for constructing a sparse affinity sub-matrix (Huang et al. 2019).

Other approaches use the properties of the small initial clusters for the affinity computation. Clustering Based on Graph of Intensity Topology (GIT) estimates for example a global topological graph (topo-graph) between local clusters (Gao et al. 2021). It then uses the Wasserstein Distance between predicted and prior class proportions to automatically cut noisy edges in the topo-graph and merge connected local clusters into final clusters.

The issue of characterizing the affinity between two clusters to create an edge weight is central to the efficiency of a spectral clustering algorithm operating from a submatrix.

Notice that the clustering robustness of many Spectral clustering algorithms heavily relies on the proper selection of kernel parameter, which is difficult to find without prior knowledge (Ng, Jordan, and Weiss 2001).

# 3 Spectral Bridges

The proposed algorithm uses k-means centroids for vector quantization defining Voronoï region, and a strategy is proposed to link these regions, with an "affinity" gauged in terms of minimal margin between pairs of classes. These affinities are considered as weight of edges defining a completely connected graph whose vertices are the regions. Spectral clustering on the region provide a partition of the input space. The sole parameters of the algorithm are the number of Voronoï region and the number of final cluster.

## 3.1 Bridge affinity

The basic idea involves calculating the difference in inertia achieved by projecting onto a segment connecting two centroids, rather than using the two centroids separately (see Figure 1). If the difference is small, it suggests a low density between the classes. Conversely, if this diffrence is large, it indicates that the two classes may reside within the same densely populated region.

Let us consider a sample $X = (\boldsymbol{x}_i)_{i \in \{1, \cdots, n\}}$ of vectors $\boldsymbol{x}_i \in \mathbb{R}^d$ and a set of $m$ coding vectors $(\boldsymbol{\mu}_k)_{k \in \{1, \cdots, m\}}$ defining a partition $P = \{\mathcal{V}_1, \cdots, \mathcal{V}_m\}$ of $\mathbb{R}^d$ into $m$ Voronoï regions:

$$\mathcal{V}_k = \left\{ \mathbf{x} \in \mathbb{R}^d \mid \|\mathbf{x} - \boldsymbol{\mu}_k\| \leq \|\mathbf{x} - \boldsymbol{\mu}_j\| \text{ for all } j \neq k \right\}.$$

In the following a ball denotes the subset of $X$ in a Voronoï region. The inertia of two balls $\mathcal{V}_k$ and $\mathcal{V}_l$ is

$$I_{kl} = \sum_{\boldsymbol{x}_i \in \mathcal{V}_k} \|\boldsymbol{x}_i - \boldsymbol{\mu}_k\|^2 + \sum_{\boldsymbol{x}_i \in \mathcal{V}_l} \|\boldsymbol{x}_i - \boldsymbol{\mu}_l\|^2.$$

We define a bridge as a structure defined by a segment connecting two centroids $\boldsymbol{\mu}_k$ and $\boldsymbol{\mu}_l$. The inertia of a bridge between $\mathcal{V}_k$ and $\mathcal{V}_l$ is defined as

$$B_{kl} = \sum_{\boldsymbol{x}_i \in \mathcal{V}_k \cup \mathcal{V}_l} \|\boldsymbol{x}_i - \boldsymbol{p}_{kl}(\boldsymbol{x}_i)\|^2,$$

where

$$\boldsymbol{p}_{kl}(\boldsymbol{x}_i) = \boldsymbol{\mu}_k + t_i(\boldsymbol{\mu}_l - \boldsymbol{\mu}_k),$$

with

$$t_i = \min\left(1, \max\left(0, \frac{\langle \boldsymbol{x}_i - \boldsymbol{\mu}_k | \boldsymbol{\mu}_l - \boldsymbol{\mu}_k \rangle}{\|\boldsymbol{\mu}_l - \boldsymbol{\mu}_k\|^2}\right)\right).$$

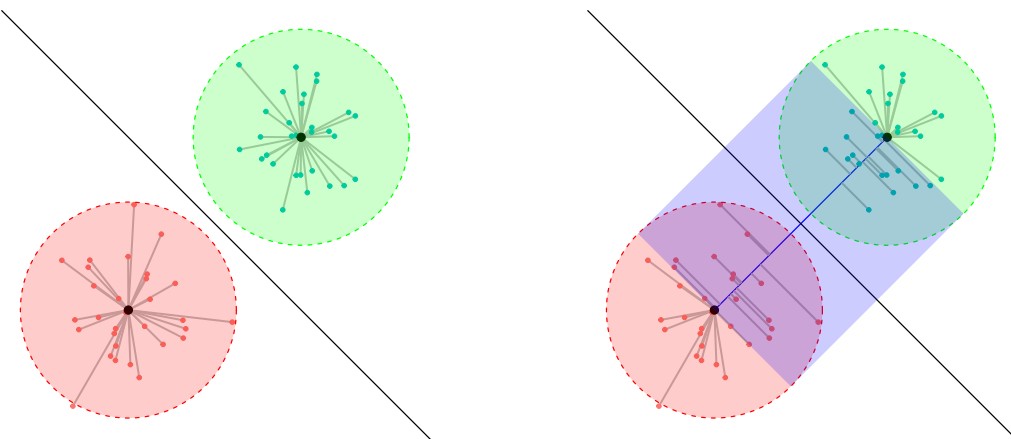

Figure 1: Balls (left) versus Bridge (right). The inertia of each structure is the sum of the squared distances represented by grey lines.

Considering two centroïds, the normalized average of the difference betweenn Bridge and balls inertia (See Appendix) constitutes the basis of our affinity measure between two regions:

$$\frac{B_{kl} - I_{kl}}{(n_k + n_l)\|\mu_k - \mu_l\|^2} = \frac{\sum_{x_i \in \mathcal{V}_k}\langle x_i - \mu_k | \mu_l - \mu_k\rangle_+^2 \sum_{x_i \in \mathcal{V}_l}\langle x_i - \mu_l | \mu_k - \mu_l\rangle_+^2}{(n_k + n_l)\|\mu_k - \mu_l\|^4},$$

$$= \frac{\sum_{x_i \in \mathcal{V}_k \cup \mathcal{V}_l} \alpha_i^2}{n_k + n_l},$$

where

$$\alpha_i = \begin{cases} t_i, & \text{if } t_i \in [0, 1/2], \\ 1 - t_i, & \text{if } t_i \in ]1/2, 1]. \end{cases}$$

The basic intuition behind this affinity is that $t_i$ represents the relative position of the projection of $x_i$ on the segment $[\mu_k, \mu_l]$. $\alpha_i$ represents the relative position on the segment, with the centroid of the class to which $x_i$ belongs as the reference point.

The boundary that separates the two clusters defined by centroids $\mu_k$ and $\mu_l$ is a hyperplane. This hyperplane is orthogonal to the line segment connecting the centroids and intersects this segment at its midpoint.

If we consider all points $x_i \in \mathcal{V}_k \cup \mathcal{V}_l$ which are not projected on centroids but somewhere on the segment, the distance from a point to the hyperplane is

$$\|p_{kl}(x_i) - \mu_{kl}\| = (1/2 - \alpha_i)\|\mu_k - \mu_l\|.$$

This distance is similar to the concept of margin in Support Vector Machine (Cortes and Vapnik 1995). When the $\alpha_i$ values are small (close to zero since $\alpha_i \in [0, 1/2]$), the margins to the hyperplane are large, indicating a low density between the classes. Conversely, if the margins are small, it suggests that the two classes may reside within the same densely populated region. Consequently, the sum of the $\alpha_i$ or $\alpha_i^2$ increases with the density of the region between the classes.

144 Note that the criterion is local and indicates the relative difference in densities between the balls and
145 the bridge, rather than evaluating a global score for the densities of the structures.

146 Eventually, we define the bridge affinity between centroids $k$ and $l$ as:

$$a_{kl} = \begin{cases} 0, & \text{if } k = l, \\ \frac{\sum_{\mathbf{x_i} \in \mathcal{V}_k \cup \mathcal{V}_l} \alpha_i^2}{n_k + n_l}, & \text{otherwise.} \end{cases}$$

147 To allow points with large margin to dominate and make the algorithm more robust to noise and
148 outliers we consider the following exponential transformation:

$$\tilde{a}_{kl} = g(a_{kl}) = \exp(\gamma \sqrt{a_{kl}}).$$

149 where $\gamma$ is a scaling factor. This factor is set to ensure a large enough separation between the final
150 coefficients. This factor is determined by the equation:

$$\gamma = \frac{log(M)}{\sqrt{q_{90}} - \sqrt{q_{10}}}$$

151 where $q_{10}$ and $q_{90}$ are respectively the 10th and 90th percentiles of the original affinity matrix
152 and $M > 1$. Thus, since the transformation is order-preserving, the 90th percentile of the newly
153 constructed matrix is $M$ times greater than the 10th percentile. By default, $M$ is arbitrarily set to a
154 large value of $10^4$.

155 The inclusion of the square root can be understood as redefining the affinity measure. Instead of
156 considering the variance and the squared Euclidean norm, we interpret the affinity as the ratio
157 between the standard deviation and the length of the segment connecting two centroids. This
158 reinterpretation greatly enhances numerical stability, contributing to more reliable clustering results.

## 3.2 Algorithm

160 The Spectral Bridges algorithm first identifies local clusters to define Voronoï regions, computes
161 edges with affinity weights between these regions, and ultimately cuts edges between regions with
162 low inter-region density to determine the final clusters (See Algorithm 1 and Figure 2).

163 In spectral clustering, the time complexity is usually dominated by the eigen-decomposition step,
164 which is $O(n^3)$. However, in the case of Spectral Bridges, the k-means algorithm has a time complexity
165 of $O(n \times m \times d)$. For datasets with large $n$, this can be more significant than the $O(m^3)$ time complexity
166 of the Spectral Bridges eigen-decomposition. As for the affinity matrix construction, there are $m^2$
167 coefficients to be calculated. Each $a_{kl}$ coefficient requires the computation of $n_k + n_l$ dot products as
168 well as the norm $\|\mu_k - \mu_l\|$, the latter often being negligeable. Assuming that the Voronoï regions are
169 roughly balanced in cardinality, we have $n_k \approx \frac{n}{m}$. Since $m$ should always be less than $n$, therefore
170 $\frac{n}{m} > 1$ and the time complexity of the affinity matrix is $O(\frac{n}{m} \times m^2 \times d) = O(n \times m \times d)$ given the
171 acceptable range of values for $m$. Nonetheless, this is rarely the bottleneck.

---

**Algorithm 1** Spectral Bridges

---

1: **procedure** SPECTRALBRIDGES($X, k, m$) ▷ $X$: input dataset, $k$: number of clusters, $m$: number of Voronoï regions
2:     **Step 1: Vector Quantization**
3:     centroids, voronoiRegions ← KMEANS($X, m$) ▷ Initial centroids and Voronoi regions using k-means++
4:     **Step 2: Affinity Computation**
5:     $A = \{g(a_{kl})\}_{kl}$ ← AFFINITY($X$, centroids, voronoiRegions) ▷ Compute affinity matrix $A$
6:     **Step 3: Spectral Clustering** ▷ Assign each region to a cluster
7:     labels ← SPECTRALCLUSTERING($A, k$)
8:     **Step 4: Propagate** ▷ Assign each data point to the cluster of its region
9:     clusters ← PROPAGATE($X$, labels, voronoiRegions)
10:    **return** clusters ▷ Return cluster labels for data points in $X$
11: **end procedure**

---

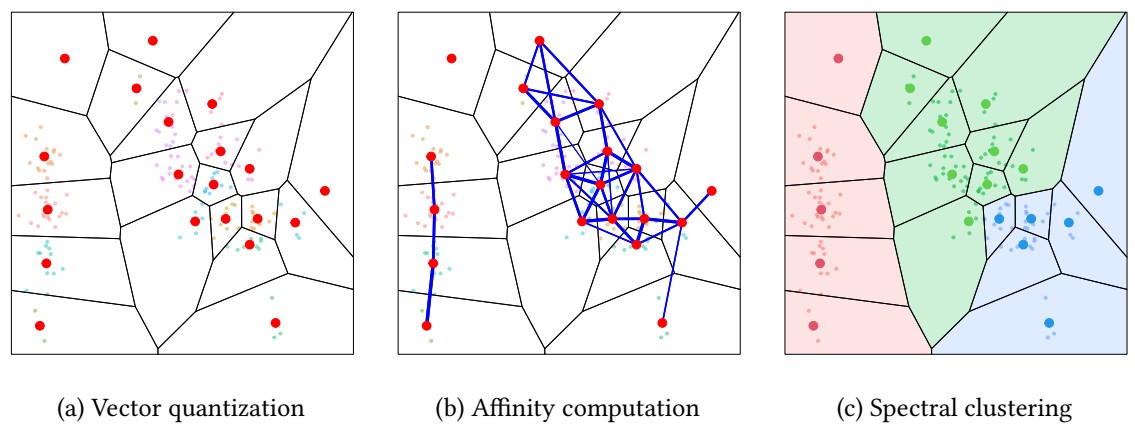

(a) Vector quantization      (b) Affinity computation      (c) Spectral clustering

Figure 2: Illustration of the Spectral bridges algorithm with the Iris dataset (first principal plane). Vector quantization (Step 1 of Algorithm 1 ), Affinity computation (Step 2 of Algorithm 1 ), Spectral clustering and spreading (Step 3-4 of Algorithm 1 ).

## 4   Numerical experiments

In this section, the results obtained from testing the Spectral Bridges algorithm on various datasets, both small and large scale, including real-world and well-known synthetic datasets, are presented. These experiments assess the accuracy, time and space complexity, ease of use, robustness, and adaptability of our algorithm. We compare Spectral Bridges (SB) against several state-of-the-art methods, including k-means++ (KM) (MacQueen et al. 1967; Arthur and Vassilvitskii 2006), Expectation-Maximization (EM) (Dempster, Laird, and Rubin 1977), Ward Clustering (WC) (Ward Jr 1963), and DBSCAN (DB) (Ester et al. 1996). This comparison establishes baselines across centroid-based clustering algorithms, hierarchical methods, and density-based methods.

The algorithms are evaluated on both raw and PCA-processed data with varying dimensionality. For synthetic datasets, Gaussian and/or uniform noise is introduced to assess the robustness of the algorithm.

## 4.1 Datasets

### 4.1.1 Real-world data

- **MNIST**: A large dataset containing 60,000 handwritten digit images in ten balanced classes, commonly used for image processing benchmarks. Each image consists of $28 \times 28 = 784$ pixels.
- **UCI ML Breast Cancer Wisconsin**: A dataset featuring computed attributes from digitized images of fine needle aspirates (FNA) of breast masses, used to predict whether a tumor is malignant or benign.

### 4.1.2 Synthetic data

- **Impossible**: A synthetic dataset designed to challenge clustering algorithms with complex patterns.
- **Moons**: A two-dimensional dataset with two interleaving half-circles.
- **Circles**: A synthetic dataset of points arranged in two non-linearly separable circles.
- **Smile**: A synthetic dataset with points arranged in the shape of a smiling face, used to test the separation of non-linearly separable data.

### 4.1.3 Datasets Summary & Class Balance

Table 1: Datasets Summary & Class Balance

| Dataset | #Dims | #Samples | #Classes | Class Proportions |
|---------|-------|----------|----------|-------------------|
| MNIST | 784 | 60000 | 10 | 9.9%, 11.2%, 9.9%, 10.3%, 9.7%, 9%, 9.9%, 10.4%, 9.7%, 9.9% |
| Breast Cancer | 30 | 569 | 2 | 37.3%, 62.7% |
| Impossible | 2 | 3594 | 7 | 24.8%, 18.8%, 11.3%, 7.5%, 12.5%, 12.5%, 12.5% |
| Moons | 2 | 1000 | 2 | 50%, 50% |
| Circles | 2 | 1000 | 2 | 50%, 50% |
| Smile | 2 | 1000 | 4 | 25%, 25%, 25%, 25% |

Class proportions are presented in ascending order starting from label 0.

## 4.2 Metrics

To evaluate the performance of the clustering algorithm, the Adjusted Rand Index (ARI) (Halkidi, Batistakis, and Vazirgiannis 2002) and Normalized Mutual Information (NMI) (Cover and Thomas 1991) are used. ARI measures the similarity between two clustering results, ranging from -0.5 to 1, with 1 indicating perfect agreement. NMI ranges from 0 to 1, with higher values indicating better clustering quality. In some tests, the variability of scores across multiple runs is also reported due to the random initialization in k-means, though k-means++ generally provides stable and reproducible results.

## 4.3 Platform

All experiments were conducted on an Archlinux machine with Linux 6.9.3 Kernel, 8GB of RAM, and an AMD Ryzen 3 7320U processor.

## 4.4 Hyperparameter settings

The hyperparameters of the Spectral Bridges algorithm were based on the size of each dataset, $n$, and the number of clusters, $K$. A larger number of clusters typically suggests that a higher value for the number of Voronoï regions is optimal. Conversely, using a high number of Voronoï regions for a small dataset might result in nearly empty regions that do not adequately represent any local structure.

A good yet not very precise way of setting the number of Voronoï regions $m$ is to observe the Within Cluster Sum of Squares (WCSS) or inertia in a way akin to the elbow method. Since $m$ should be set to a value strictly greater than $K$, we plot the WCSS for varying values of $m$, and find a value such that the WCSS-$m$ relationship becomes quasi-linear.

By adjusting $m$ in this manner, we aim to balance the need for detailed representation with the risk of overfitting, ensuring that each Voronoï region meaningfully captures the underlying data distribution. The sensitivity or lack thereof is illustrated later on by Figure 10.

For other algorithms, such as DBSCAN, labels were used to determine the best hyperparameter values to compare our method against the "best case scenario", thus putting the Spectral Bridges algorithm at a voluntary disadvantage.

## 4.5 Time complexity

To assess the algorithm's time complexity, the average execution times over 50 runs were computed for varying numbers of Voronoï regions $m$ as well as dataset sizes. With a constant number of clusters $K = 5$ and an embedding dimension of $d = 10$, the results (see Figure 3) highlight Spectral Bridges algorihtm's efficacy. As discussed previously, we observe a linear relationship between $m$ and the execution time because the matrix construction is highly optimized and the time taken is almost negligeable compared to that of the initial k-means++ centroids initalization.

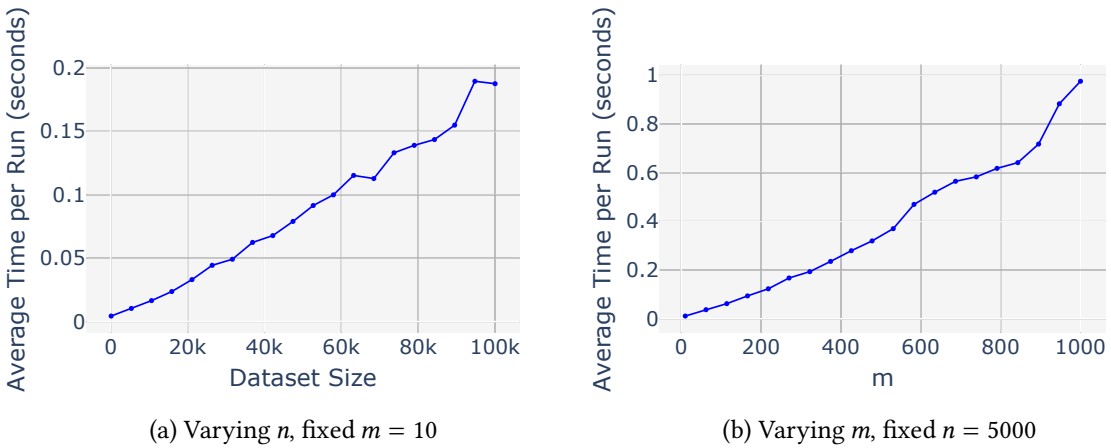

(a) Varying $n$, fixed $m = 10$        (b) Varying $m$, fixed $n = 5000$

Figure 3: Average time taken per model fit.

## 4.6 Accuracy

The algorithm's accuracy was first evaluated on the MNIST dataset. Metrics were collected to compare our method with k-means++, EM, and Ward clustering. Metric were estimated by taking the empirical average over 10 consecutive runs with the same random seed for each method. Since our computational capabilites were too limited, a sample of 20,000 (one third) data points was chosen at random for each iteration.

Let $h$ denote the embedding dimension of the dataset. Spectral Bridges was tested both on the raw MNIST dataset without preprocessing ($h = 784$) and after reducing its dimension using PCA to $h \in \{8, 16, 32, 64\}$ (see Figure 4).

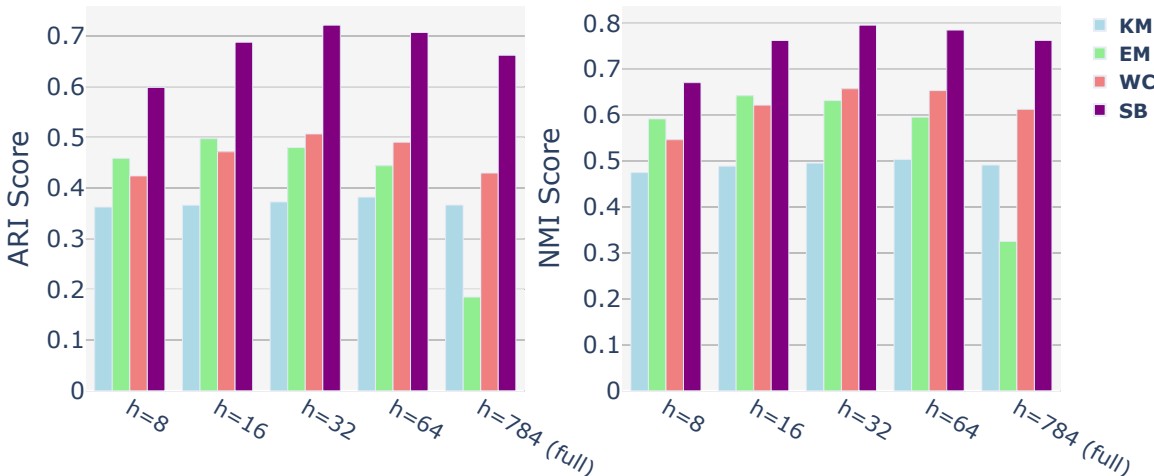

Figure 4: ARI and NMI scores of k-means++ (blue), EM (green), Ward Clustering (red), and Spectral Bridges (purple) on PCA embedding and full MNIST.

For visualization purposes, the predicted clusters by Spectral Bridges and k-means++ were projected using UMAP to compare them against the ground truth labels and to better understand the cluster shapes (see Figure 5). Note that the projection was not used in the experiments as an embedding, and thus does not play any role in the clustering process itself. As a matter of fact, the embedding used was obtained with PCA, $h = 32$ and 250 Voronoï regions. Note that the label colors match the legend only in the case of the ground truth data. Indeed, the ordering of the labels have no significance on clustering quality.

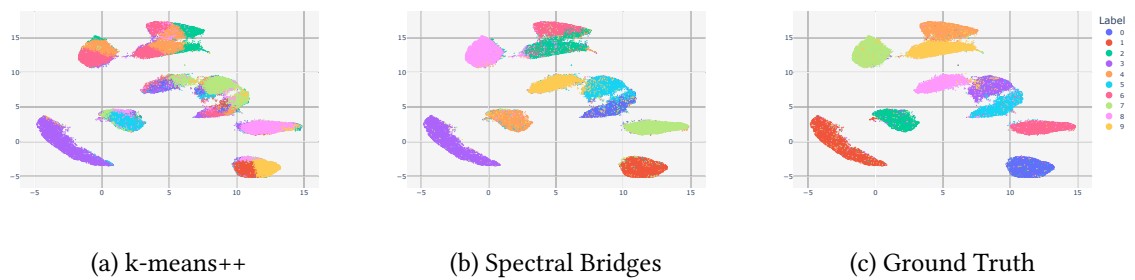

(a) k-means++ (b) Spectral Bridges (c) Ground Truth

Figure 5: UMAP projection of predicted clusters against the ground truth labels.

The Spectral Bridges algorithm was also put to the test against the same competitors using scikit-learn's UCI Breast Cancer data. Once again, this new method performs well although the advantage is not as obvious in this case (see Figure 6). However, in none of our tests has it ranked worse than k-means++. The results are displayed as a boxplot generated from 200 iterations of each algorithm using a different seed, in order to better grasp the variability lying in the seed dependent nature of the k-means++, Expectation Maximization and Spectral Bridges algorithms.

Since the Spectral Bridges algorithm is expected to excel at discerning complex and intricate cluster

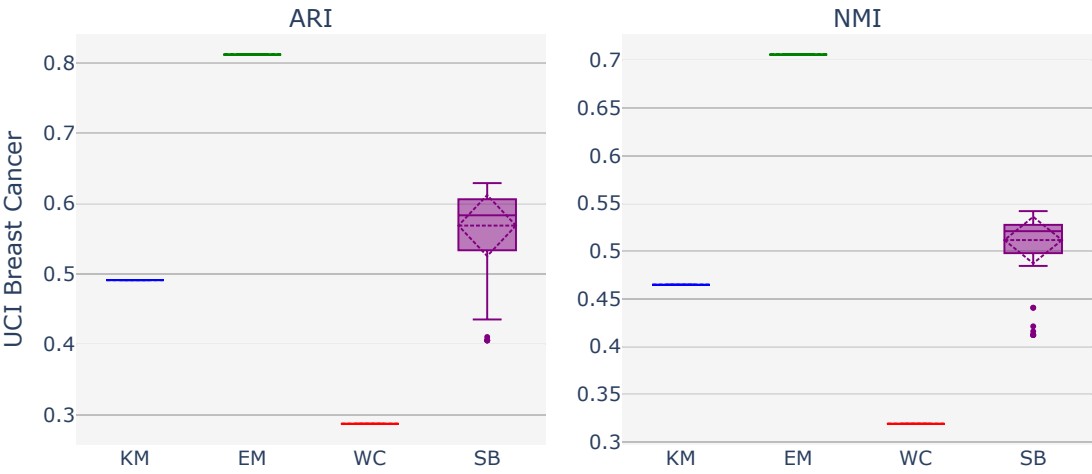

Figure 6: ARI and NMI scores of k-means++ (blue), EM (green), Ward Clustering (red), and Spectral Bridges (purple) on the UCI Breast Cancer dataset.

structures, an array of four toy datasets was collected, as illustrated in Figure 7.

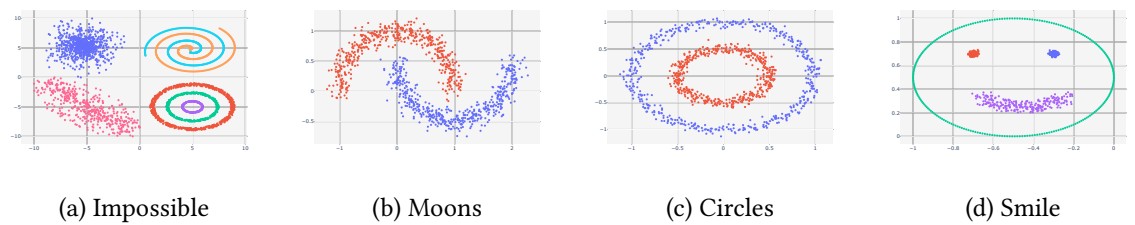

(a) Impossible      (b) Moons      (c) Circles      (d) Smile

Figure 7: Four toy datasets.

Multiple algorithms, including the proposed one, were benchmarked in the exact same manner as for the UCI Breast Cancer data. The results show that the proposed method outperforms all tested algorithms (DBSCAN, k-means++, Expectation Maximization, and Ward Clustering) while requiring few hyperparameters. As previously discussed, DBSCAN's parameters were optimized using the ground truth labels to represent a best-case scenario; however, in practical applications, suboptimal performance is more likely. Despite this optimization, the Spectral-Bridge algorithm still demonstrates superior ability to capture and represent the underlying cluster structures.

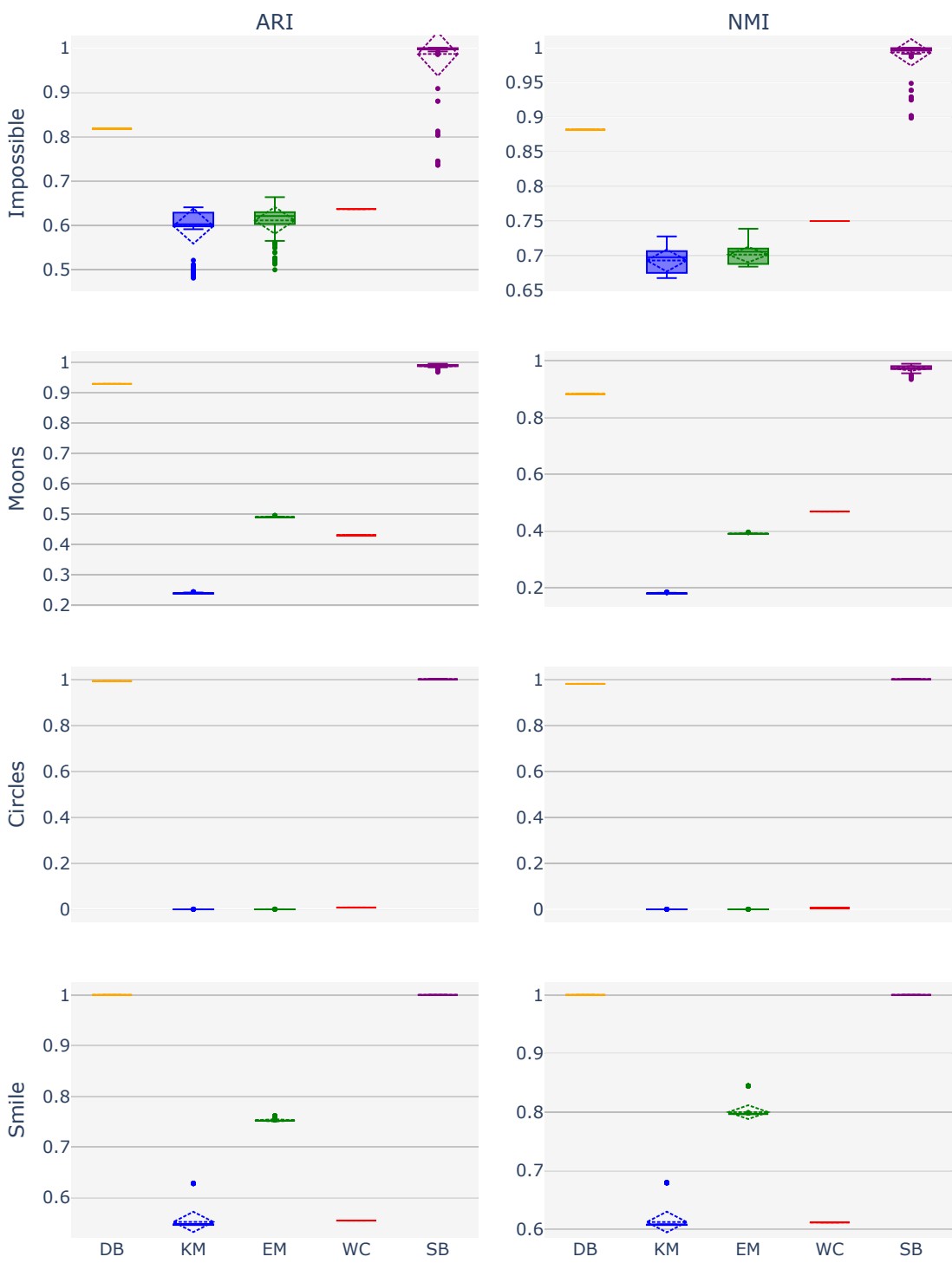

Figure 8: ARI and NMI scores of Spectral Bridges and competitors on standard synthetic toy datasets.

## 4.7 Noise robustness

To evaluate the noise robustness of the algorithm, two experimental setups were devised: one involved introducing Gaussian-distributed perturbations to the data, and the other involved concatenating uniformly distributed points within a predefined rectangular region (determined by the span of the dataset) to the existing dataset. As illustrated in Figure 9, the tests demonstrate that in both scenarios, the algorithm exhibits a high degree of insensitivity to noise.

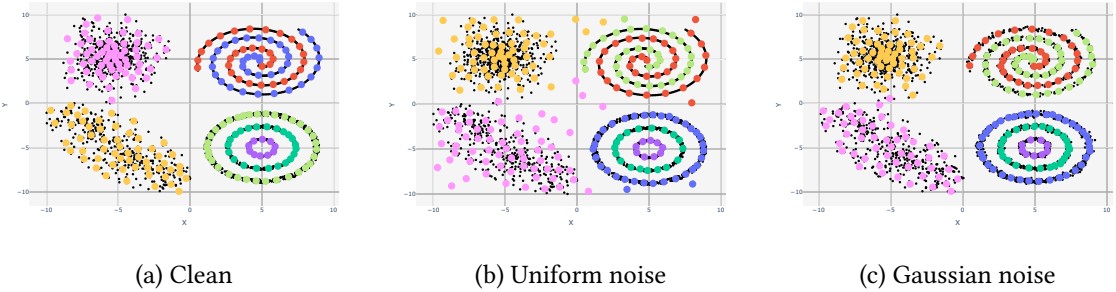

(a) Clean  (b) Uniform noise  (c) Gaussian noise

Figure 9: Three representations of the algorithm's predicted cluster centers are displayed as colored dots, with each point of the Impossible dataset shown as a small black dot. In the left graph, the dataset is unmodified. In the center graph, 250 uniformly distributed samples were added. In the right graph, Gaussian noise perturbations with $\sigma = 0.1$ were applied.

## 4.8 Hyperparameter values effect on accuracy

To better understand and measure the significance of choosing the right values for the hyper-parameters of the proposed algorithm, that it to say the number of Voronoï regions $m$, Spectral Bridges was run on the PCA $h = 32$ embedded MNIST dataset with varying values of $m \in \{10, 120, 230, 340, 450, 560, 670, 780, 890, 1000\}$. The case $m = 10$ is equivalent to the k-means++ algorithm. ARI and NMI scores are recorded over 20 consecutive iterations and subsequently plotted. As shown by Figure 10, the accuracy seems to be consistently increasing with values of $m$, although the largest observed gap occurs between values of $m = 10$ and $m = 120$, indicating a tremendous improvement over the classical k-means++ framework even for empirically suboptimal hyperparameter values.

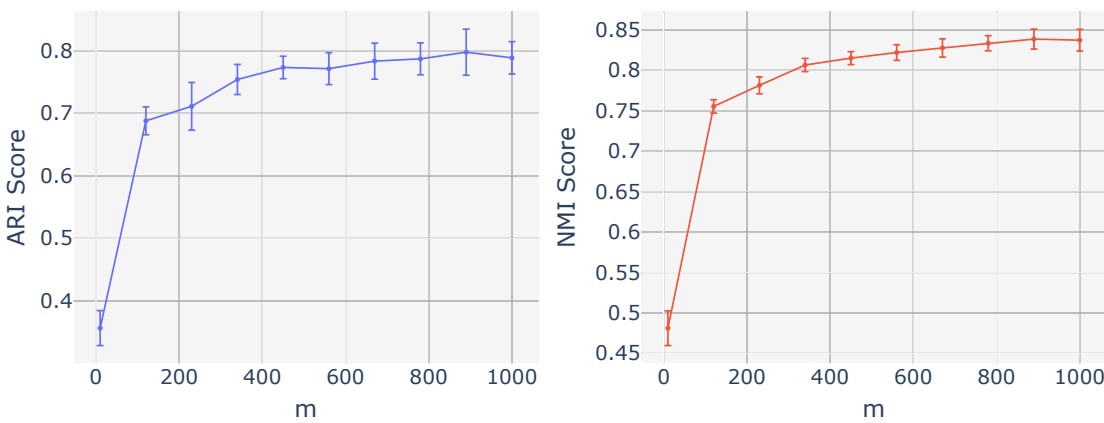

Figure 10: ARI and NMI scores of Spectral Bridges with varying values of $m$.

# 5    Conclusive remarks

Spectral Bridges is an original clustering algorithm which presents a novel approach by integrating the strengths of traditional k-means and spectral clustering frameworks. This algorithm utilizes a simple affinity measure for spectral clustering, which is derived from the minimal margin between pairs of Voronoï regions.

The algorithm demonstrates scalability, handling large datasets efficiently through a balanced computational complexity between the k-means clustering and eigen-decomposition steps. As a non-parametric method, Spectral Bridges does not rely on strong assumptions about data distribution, enhancing its versatility across various data types. It performs exceptionally well with both synthetic and real-world data and consistently outperforms conventional clustering algorithms such as k-means, DBSCAN, and mixture models.

The design of Spectral Bridges ensures robustness to noise, a significant advantage in real-world applications. Additionally, the algorithm requires minimal hyperparameters, primarily the number of Voronoï regions, making it straightforward to tune and deploy.

Furthermore, Spectral Bridges can be kernelized, allowing it to handle data in similarity space directly, which enhances its flexibility and applicability. Overall, Spectral Bridges is a powerful, robust, and scalable clustering algorithm that offers significant improvements over traditional methods, making it an excellent tool for advanced clustering tasks across numerous domains.

# 6    Appendix

## 6.1    Derivation of the bridge affinity

We denote a bridge as a segment connecting two centroids $\mu_k$ and $\mu_l$. The inertia of a bridge between $\mathscr{V}_k$ and $\mathscr{V}_l$ is defined as

$$B_{kl} = \sum_{\boldsymbol{x}_i \in \mathscr{V}_k \cup \mathscr{V}_l} \|\boldsymbol{x}_i - \boldsymbol{p}_{kl}(\boldsymbol{x}_i)\|^2,$$

where

$$\boldsymbol{p}_{kl}(\boldsymbol{x}_i) = \boldsymbol{\mu}_k + t_i(\boldsymbol{\mu}_l - \boldsymbol{\mu}_k),$$

with

$$t_i = \min\left(1, \max\left(0, \frac{\langle \boldsymbol{x}_i - \boldsymbol{\mu}_k | \boldsymbol{\mu}_l - \boldsymbol{\mu}_k \rangle}{\|\boldsymbol{\mu}_l - \boldsymbol{\mu}_k\|^2}\right)\right).$$

$B_{kl}$, the bridge inertia between centroids $k$ and $l$, can be expressed as the sum of three terms, which represents the projection onto each centroïds and onto the segment:

$$B_{kl} = \sum_{i|t_i=0} \|\boldsymbol{x}_i - \boldsymbol{\mu}_k\|^2 + \sum_{i|t_i=1} \|\boldsymbol{x}_i - \boldsymbol{\mu}_l\|^2 + \sum_{i|t_i\in]0,1[} \|\boldsymbol{x}_i - \boldsymbol{p}_{kl}(\boldsymbol{x}_i)\|^2.$$

The last term may be decomposed in two parts corresponding to the points of the two Voronoï regions which are projected on the segment:

$$\sum_{i|t_i\in]0,1[} \|\boldsymbol{x}_i - \boldsymbol{p}_{kl}(\boldsymbol{x}_i)\|^2 = \sum_{i|t_i\in]0,\frac{1}{2}[} \|\boldsymbol{x}_i - \boldsymbol{p}_{kl}(\boldsymbol{x}_i)\|^2 + \sum_{i|t_i\in[\frac{1}{2},1[} \|\boldsymbol{x}_i - \boldsymbol{p}_{kl}(\boldsymbol{x}_i)\|^2$$

309 and each part further decomposed using Pythagore

$$\sum_{i|t_i\in]0,\frac{1}{2}[} \|\boldsymbol{x}_i - \boldsymbol{p}_{kl}(\boldsymbol{x}_i)\|^2 = \sum_{i|t_i\in]0,\frac{1}{2}[} \|\boldsymbol{x}_i - \boldsymbol{\mu}_k\|^2 - \sum_{i|t_i\in]0,\frac{1}{2}[} \|\boldsymbol{\mu}_k - \boldsymbol{p}_{kl}(\boldsymbol{x}_i)\|^2$$

$$= \sum_{i|t_i\in]0,\frac{1}{2}[} \|\boldsymbol{x}_i - \boldsymbol{\mu}_k\|^2 - \sum_{i|t_i\in]0,\frac{1}{2}[} \|t_i(\boldsymbol{\mu}_k - \boldsymbol{\mu}_l)\|^2,$$

$$\sum_{i|t_i\in]\frac{1}{2},1[} \|\boldsymbol{x}_i - \boldsymbol{p}_{kl}(\boldsymbol{x}_i)\|^2 = \sum_{i|t_i\in]0,\frac{1}{2}[} \|\boldsymbol{x}_i - \boldsymbol{\mu}_l\|^2 - \sum_{i|t_i\in]0,\frac{1}{2}[} \|\boldsymbol{\mu}_l - \boldsymbol{p}_{kl}(\boldsymbol{x}_i)\|^2$$

$$= \sum_{i|t_i\in]\frac{1}{2},1[} \|\boldsymbol{x}_i - \boldsymbol{\mu}_k\|^2 - \sum_{i|t_i\in]0,\frac{1}{2}[} \|(1 - t_i)(\boldsymbol{\mu}_k - \boldsymbol{\mu}_l)\|^2$$

310 Thus

$$B_{kl} - I_{kl} = \sum_{i|t_i\in]0,\frac{1}{2}[} t_i^2\|\boldsymbol{\mu}_k - \boldsymbol{\mu}_l\|^2 + \sum_{i|t_i\in]\frac{1}{2},1[} (1 - t_i)^2\|\boldsymbol{\mu}_k - \boldsymbol{\mu}_l\|^2,$$

$$\frac{B_{kl} - I_{kl}}{\|\boldsymbol{\mu}_k - \boldsymbol{\mu}_l\|^2} = \sum_{i|t_i\in]0,\frac{1}{2}[} t_i^2 + \sum_{i|t_i\in]\frac{1}{2},1[} (1 - t_i)^2,$$

$$\frac{B_{kl} - I_{kl}}{(n_k + n_l)\|\boldsymbol{\mu}_k - \boldsymbol{\mu}_l\|^2} = \frac{\sum_{\boldsymbol{x_i}\in\mathscr{V}_k}\langle\boldsymbol{x_i} - \boldsymbol{\mu}_k|\boldsymbol{\mu}_l - \boldsymbol{\mu}_k\rangle_+^2 + \sum_{\boldsymbol{x_i}\in\mathscr{V}_l}\langle\boldsymbol{x_i} - \boldsymbol{\mu}_l|\boldsymbol{\mu}_k - \boldsymbol{\mu}_l\rangle_+^2}{(n_k + n_l)\|\boldsymbol{\mu}_k - \boldsymbol{\mu}_l\|^4}.$$

## 6.2 Code

### 6.2.1 Implementation

313 Numerical experiments have been conducted in Python. The python scripts to reproduce the
314 simulations and figures are available at https://github.com/flheight/Spectral-Bridges. The Spectral
315 Bridge algorithm is implemented both in

316 • Python: https://pypi.org/project/spectral-bridges, and
317 • R: https://github.com/cambroise/spectral-bridges-Rpackage.

### 6.2.2 Affinity matrix computation

319 Taking a closer look at the second step of Algorithm 1 , that is the affinity matrix calculation
320 with a $O(n \times m \times d)$ time complexity, most operations can be parallelized leaving a single loop,
321 bundling together $m^2$ dot products into only $m$ matrix multiplications, thus allowing for an efficient
322 construction in both high and low level programming languages. Though the complexity of the
323 algorithm remains unchanged, libraries such as Basic Linear Algebra Subprograms can render the
324 calculations orders of magnitude faster. Moreover, the symmetrical nature of the bridge affinity can
325 be used to effectively halve the computation time.

326 The calculation of the affinity matrix is highlighted by the Python code Listing 1. Though it could
327 be even more optimized, the following code snippet is approximately 200 times faster than a naive
328 implementation on a small dataset comprised of $n = 3594$, $d = 2$ points, and a value of $m = 250$.

329 Notice that the Python code is significantly faster than the R code.

**Listing 1** Python code for affinity matrix computation

```python
# Initialize the affinity matrix
affinity = np.empty((self.n_nodes, self.n_nodes))

# Center each Voronoi region around its centroid
X_centered = [
    X[kmeans.labels_ == i] - kmeans.cluster_centers_[i] for i in range(self.n_nodes)
]

# Count the total number of points in each pair of regions
counts = np.array([X_centered[i].shape[0] for i in range(self.n_nodes)])
counts = counts[np.newaxis, :] + counts[:, np.newaxis]

# Compute the segments between each pair of centroids and their squared Euclidean norm
segments = (
    kmeans.cluster_centers_[np.newaxis, :] - kmeans.cluster_centers_[:, np.newaxis]
)
dists = np.einsum("ijk,ijk->ij", segments, segments)
np.fill_diagonal(dists, 1)  # Avoid dividing by zero

# Assign each row of the affinity matrix
for i in range(self.n_nodes):
    projs = np.maximum(np.dot(X_centered[i], segments[i].T), 0)
    affinity[i] = np.einsum("ij,ij->j", projs, projs)

# Symmetrize the matrix and normalize, as well as taking the element-wise square root
affinity = np.sqrt(affinity + affinity.T) / (np.sqrt(counts) * dists)
affinity -= 0.5 * affinity.max()  # For numerical stability

# Apply the exponential transformation
q10, q90 = np.quantile(affinity, [0.1, 0.9])

gamma = np.log(self.M) / (q90 - q10)
affinity = np.exp(gamma * affinity)
```

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

# Session information

```
R version 4.3.2 (2023-10-31)
Platform: x86_64-apple-darwin20 (64-bit)
Running under: macOS Sonoma 14.3.1

Matrix products: default
BLAS:   /Library/Frameworks/R.framework/Versions/4.3-x86_64/Resources/lib/libRblas.0.dylib
LAPACK: /Library/Frameworks/R.framework/Versions/4.3-x86_64/Resources/lib/libRlapack.dylib;  LAPAC

locale:
```

```
[1] en_US.UTF-8/en_US.UTF-8/en_US.UTF-8/C/en_US.UTF-8/en_US.UTF-8

time zone: Europe/Paris
tzcode source: internal

attached base packages:
[1] stats     graphics  grDevices utils     datasets  methods   base

loaded via a namespace (and not attached):
 [1] digest_0.6.34    fastmap_1.1.1      xfun_0.42       Matrix_1.6-5
 [5] lattice_0.22-5   reticulate_1.37.0 knitr_1.45      htmltools_0.5.7
 [9] png_0.1-8        rmarkdown_2.26    cli_3.6.2       grid_4.3.2
[13] compiler_4.3.2   rstudioapi_0.15.0 tools_4.3.2     evaluate_0.23
[17] Rcpp_1.0.12      yaml_2.3.8        rlang_1.1.3     jsonlite_1.8.8
```

