# OpenReview forum: "Spectral Bridges"
_Computo — Accepted by Computo_

### Review · Reviewer_4dJq · 2024-09-23

**Summary Of Contributions:**

The authors propose a new clustering algorithm called spectral bridge. The method consists in first performing a pre-clustering of the data in $m$ clusters based on the k-means algorithms, with $m$ larger than the final number of clusters $k$. The second step aggregates the $m$ pre-clusters in $k$ clusters. This second step performs a spectral clustering based on the affinity matrix computed on the pre-clusters, this matrix is calculated based on the so-called bridge affinity between each pair of pre-clusters. This bridge affinity between two pre-clusters is based on the difference between the inertia obtained by projecting the points of the two pre-clusters onto a segment connecting two centroids and the usual intra-class inertia—a small value of this difference being related to a low density between the classes. The final partitioning of the space in $k$ clusters is obtained by merging the Voronoï regions that were defined by the $m$ pre-clusters.

The article is composed of 6 sections. Section 1 gives an introduction to the article. Section 2 gives an overview of the related works, especially it insists on the intractability of spectral clustering when the number $n$ of data is large and gives other references based on small initial clusters. Section 3 details the approach proposed by the authors. Section 4 presents the numerical experiments on two real-world datasets (MNIST and UCI ML Breast Cancer Wisconsin) and synthetic datasets. The Spectral Bridges method is compared with k-means++, mixture of Gaussians, Ward Clustering and DBSCAN. Section 5 presents conclusive remarks discussing all the advantages of the proposed approach.  Section 6 in the Appendix presents the derivation of the bridge affinity discusses numerical implementation and gives Python code for the affinity matrix computation.

**Audience:**

Yes

**Broader Impact Concerns:**

I have no remarks on this part.

**Claims And Evidence:**

Yes

**Requested Changes:**

In the introduction, the authors should mention the auto-encoders (AE) and variational auto-encoder (VAE) approaches which are quite efficient for embedding the data in a low dimensional space where the clustering is easier. In Section 3.1 p.4 l.118 the authors should give further intuition to explain why the bridge affinity is small in low-density areas. For instance, the authors could also compare their methods to the article Agarap, A. F., & Azcarraga, A. P. (2020, July). *Improving k-means clustering performance with disentangled internal representations*. In 2020 *International Joint Conference on Neural Networks (IJCNN)* (pp. 1-8). IEEE.

There is an error in the formula p.5 l.129 where there should only be a  $+$ symbol between the two sums. (Same typo in the Appendix).

The article should also discuss the choice of the number of clusters.

The numerical experiments should not only include basic competitors but should also include cutting-edge ones such as clustering based on Graph of Intensity Topology (GIT) presented in the related work part, the standard spectral clustering approach, the ultra-scalable spectral clustering (U-SPEC), and also approaches based on AE or VAE.

From the dataset considered point of view, the dimension and the number of variables considered are relatively low, I recommend the authors to find additional real datasets to allow a more complete comparison.

p.9 l. 236: DBSCAN is missing from the list.

p.9 l.237: the term " same random seed for each method " is not very explicit. Is this related to a random initial centre that would be the same across methods? Otherwise is it important that different methods share the same seed.

P.9 l.239: "20,000 (one-third) data points were chosen at random for each iteration" should also be detailed. Why not run the algorithm only on the same 20,000 data points? Sampling the point at each iteration of the algorithm is not so standard from a clustering point of view and could have consequences on the final result.

In Figure 5 p.10: The authors use the UMAP projection to visualize the results. In the Figure, the ground truth results are relatively well separated. Thus, would it be relevant to directly perform the clustering on the UMAP projection?

In Figure 6, why do only spectral bridges show variability in the results?

p.15: The alignment of equations is odd. I recommend aligning equations on the left side.

**Strengths And Weaknesses:**

The paper is well written, it presents a novel approach for clustering which is scalable. It defines a new notion of affinity between clusters which is quite interpretable and easy to compute. Moreover, the number of tuning parameters required by the approach is relatively low (number of clusters, number of pre-clusters). The main weakness of the paper is the experimental evaluation which is performed on not enough real datasets and omitting some important competitors. Moreover, from the reproducibility point of view, I was not able to regenerate all the figures of the article by using the Make-figures.Rmd file, I was also expecting to be able to run again the numerical experiments that have generated the table in the article.

---

> ### Author Response · Authors · 2024-11-15
> **Auto-Encoders and VAE**
>
> Dear Reviewer,
>
> Thank you for your thoughtful review and for highlighting both the strengths and the areas for improvement in our manuscript. Your feedback is extremely valuable, and we appreciate the detailed comments you've provided. Below, we address each of your suggestions in detail.
>
> ### Auto-Encoders (AE) and Variational Auto-Encoders (VAE)
>
> We agree that AE and VAE are relevant approaches for embedding data into a lower-dimensional space, where clustering can be more efficient. Many dimension reduction methods can be used before clustering and sometimes improve the result. Although our algorithm is designed for direct use, we tested all algorithms after a UMAP dimension reduction, as UMAP is considered the state of the art in the domain.

---

> ### Author Response · Authors · 2024-11-15
> **Intuition Behind Bridge Affinity**
>
> ### Intuition Behind Bridge Affinity
>
> We appreciate your suggestion to provide further intuition regarding why the bridge affinity is small in low-density areas. We added a Figure in section 3.1 to better explain this concept. The bridge affinity is a normalized sum of the $\alpha_i^2$. The $\alpha_i$ is a function of the distance of each point to the mediating hyperplane.  The larger $\alpha_i$ the closer vector $x_i$ is from the mediating hyperplane.
>
> We added a new Figure (as mentioned in the reply to reviewer 1) to illustrate the SVM interpretation

---

> ### Author Response · Authors · 2024-11-15
> **Choice of Number of Clusters**
>
> ### Choice of Number of Clusters
>
>   The determination of the optimal number of clusters remains a challenging model selection problem. Our algorithm, being rooted in geometrical considerations rather than a statistical framework, currently lacks a robust and systematic method for selecting the appropriate number of clusters. This is an area we recognize as needing further development, and we plan to explore statistical techniques to enhance the reliability of cluster determination in future work.

---

> ### Author Response · Authors · 2024-11-15
> **Additional Competitors in Numerical Experiments and datasets**
>
> ### Additional Competitors in Numerical Experiments and datasets
>
> We appreciate your suggestion to include more advanced competitors in the numerical experiments.
>
> We added:
>
> -   the GIT method (which was available in Python),  and
>
> -    experiments after UMAP dimension reduction.Additional Real Datasets
>
> We decided not to add more datasets as we believe that our current set of numerical experiments already provides a comprehensive evaluation of the proposed method. Furthermore, we were uncertain about which additional datasets would bring significant new insights. However, we are open to suggestions for datasets that could further validate or challenge our approach.

---

> ### Author Response · Authors · 2024-11-15
> **DBSCAN and Random Seed Clarification**
>
> ### DBSCAN and Random Seed Clarification
>
> We appreciate your comment regarding the exclusion of the DBSCAN algorithm. DBSCAN was not applied to the real datasets due to its high sensitivity to the search radius and the minimum number of samples required for cluster expansion. While tuning these parameters is feasible for simpler, synthetic datasets, it becomes significantly more challenging for complex, large-scale datasets like MNIST. Although there are strategies to determine optimal hyperparameter values, we were unable to achieve meaningful clustering results with DBSCAN on such intricate data. Consequently, we decided to omit the algorithm from our reported results.

---

> ### Author Response · Authors · 2024-11-15
> **Sampling in Numerical Experiments**
>
> ### Sampling in Numerical Experiments
>
> Regarding the sampling of 20,000 data points at each iteration, we are grateful for your suggestion to make this clearer. Due to the limited computational resources available, we sub-sampled MNIST to one-third of its original size, with each algorithm trained on this sample. This process was repeated over 10 runs to compute the mean clustering performance, with the term "iteration" referring specifically to independent training runs rather than the training process of a single algorithm. For reproducibility purposes, we set a fixed random seed at the start of all scripts. However, it is important to note that this does not imply that centroids were initialized identically, as this could vary based on the implementation of each tested algorithm.
>
> In the paper we added:
>
> >Metrics were estimated by computing the empirical average over 10 consecutive runs for each method. Due to limited computational resources, we randomly selected a sample of 20,000 data points (one-third of the total) for each run, on which all algorithms were trained and tested. To ensure reproducibility, a fixed random seed was set at the beginning of all scripts. Note, however, that this does not imply centroids were initialized identically for centroid based methods, as these may vary according to the implementation of each tested algorithm."

---

> ### Author Response · Authors · 2024-11-15
> **Clustering on UMAP Projection**
>
> ### Clustering on UMAP Projection
>
> We appreciate your observation regarding the UMAP projection in Figure 5. Given the clear separation observed in the UMAP visualization, we added experiments where clustering is performed directly on the UMAP projection to assess whether this approach could yield better clustering results.

---

> ### Author Response · Authors · 2024-11-15
> **Variability in Figure 6 and equations...**
>
> ### Variability in Figure 6
>
> This variability is due to the method's sensitivity to the initialization of the pre-clustering step. We reduced this variability with our new strategy for choosing the number of centroids.
>
> ### Equation Alignment
>
> We adjusted the alignment of the equations on page 15 to ensure they are properly aligned on the left side for better readability.
>
> We hope these revisions address your comments and concerns. Thank you again for your valuable feedback, which will significantly improve the quality and impact of our work. We look forward to your feedback on the revised manuscript.
>
> Best regards,
> Félix Laplante and Christophe Ambroise

---

### Review · Reviewer_pHXm · 2024-10-21

**Summary Of Contributions:**

The paper presents a new distance-based clustering method for continuous data. This method involves two main steps: first, a summary of the observations is obtained by subdividing the data into a large number of small Voronoi regions; then, a spectral clustering method is performed based on the similarities of the Voronoi regions. The first step reduces the computational cost of the spectral clustering step. The second step considers non-convex clusters. Therefore, the method benefits from the advantages of K-means and spectral clustering while avoiding the main problems of both methods.

Numerical experiments permit a comparison of the proposed approach with the standard clustering methods.

An R code as well as Python code are provided.

**Audience:**

No

**Claims And Evidence:**

Yes

**Requested Changes:**

Criterion $B_{kl}$

An individual can have two different contributions to the criterion: either we calculate its distance to the center, or we calculate its distance to the space defined by the vector created by two centroids. The choice of the contribution of each observation is well illustrated by Figure 1. Could the authors argue this criterion further? Why don't they take into account the distance between each observation and the line defined by two centers? I don't want to make the authors change the criterion but I would like to have more arguments on their choice.


Figure 2

I like Figure 2, but I think I misunderstood it on my first reading. I suggest adding in the legend that the centroids are indicated by a red circle (with $m=20$) and that the final partition is represented by the colored cells in the figure on the right (with $K=3$). Please explain the meaning of the blue lines (it is not clear to me).


Hyperparameters

I think the method requires two hyperparameters: the number of Voronoi cells and the dimension considered by the spectral clustering. The authors only mentioned the first hyperparameter and studied its impact (see Figure 10). Could they add a discussion on choosing the dimension considered by the spectral clustering? I would appreciate to see the impact of hyperparameters on the clustering result. The authors also gave some advice on choosing $m$ ($m$ should be larger than $K$). I suggest adding some practical advice for selecting the dimension during spectral clustering (maybe some references would be enough).


Choice of $K$

I propose to add a discussion on the choice of the number of components. I understand that this is not the purpose of the article. However, some indications on the selection of this number should be given in order to be able to apply this method on real data.

Competing method

I suggest to consider, as a competing method during the numerical experiments, the spectral clustering based on the original data.


Regularization

The authors consider a regularization of $a_{k\ell}$ (see $\tilde{a}_{k\ell}$ on page 6). Is this really necessary? If so, I think $\gamma$ is also a hyperparameter. Since the purpose of this parameter is to make the clustering output more robust to outliers, is it possible to adjust it based on the percentage of outliers in the dataset (this would require giving a specific mathematical definition of outliers)?

**Strengths And Weaknesses:**

Overall, I like the proposed method. The paper is clearly written. I particularly like Figures 1 and 2 which are very didactic. The numerical experiments illustrate the advantages of the proposed method. My main concern is the claim that the method is hyperparameter-free (made in the title and in the manuscript). For example, in Section 4.4, the authors discussed the choice of the number of Voronoi cells. I do not understand why the method would be free of the hyperparameters of usual spectral clustering methods (i.e. the choice of dimension). The accompanying R package is well implemented, it is easy to use thanks to its clear vignette. Below are my comments to the authors which I hope will be useful to them.

---

> ### Author Response · Authors · 2024-11-15
> **Hyperparameters: Number of clusters and number of centroids**
>
> ###  About hyperparameters
>
> It is worth noting that while our method does involve 3 hyperparameters :
> - Number of Clusters $K$.
> - Number of Cells $m$: This is a key parameter, as it determines the initial subdivision of the data. We provide analysis and refer to the corresponding curves to illustrate the impact of different choices for this parameter.
> - Scaling Parameter: This parameter controls the scale of the affinity computation. Unlike existing methods that often require careful tuning of scaling factors, our strategy mitigates this need by employing a more robust approach, which we explain in detail compared to traditional scaling methods.
>
> 	Our results (Figure 10) show that the algorithm is not highly sensitive to the value of the number of cells. We also observed stability relative to the scaling parameter.
>
> In the abstract we wrote
>
> > This approach is characterized by minimal hyperparameters and delineation of intricate, non-convex cluster structures.
>
> We replaced it by
>
> > This approach delineates intricate, non-convex cluster structures and is robust to hyperparameter choice.
>
>
>
> ####  Choice of K
>
> Model selection in non-parametric frameworks, such as spectral clustering applied to bridge structures, presents significant challenges due to the absence of predefined model parameters and the reliance on data-driven approaches.
>
> Such approaches as Gap Statistic could be a possible approach with a fixed number of Voronoï cells. The optimal number of clusters is where the observed clustering yields the largest gap statistic.
>
>
> ####  Choice of $m$
>
>
> The choice of the number of centroids (coding vector) is a difficult problem. Typically, a balance must be struck between having enough coding vectors to capture the underlying structure of the data and avoiding having too many since the kmeans++ step is the most time-consuming.  Figure 10 shows that increasing the number of centroids on the MNIST dataset over 120 has a limited impact.
>
> We propose a heuristic for choosing the number of coding vectors, which produces satisfactory results with all tested datasets:
>
>
> > The proposed algorithm requires three input parameters: the number of clusters \$K\$, the number of Voronoï regions \$m\$, and a scaling parameter for the spectral clustering phase.
> >
> > Model selection in non-parametric settings is challenging due to the absence of predefined model parameters. It relies heavily on data-driven approaches. Metrics like the Gap Statistic [@tibshirani2001estimating] and the Laplacian eigengap [@von2007tutorial] are potential tools for hyperparameter selection.
> >
> > We propose a method for choosing the scaling parameter (see Equation @eq-scaling) that yields stable results. Selecting both \$m\$, the number of Voronoï regions, and \$K\$, the number of clusters, is difficult. We address this by adopting a heuristic: first, choose \$K\$, then determine \$m\$ using a modified Laplacian eigengap.
> >
> > If \$K\$ represents the true number of clusters, the affinity matrix resembles a graph adjacency matrix with \$K\$ connected components. This configuration is characterized by an eigengap at the \$K\$th eigenvalue. In Self-Tuning Spectral Clustering [@zelnik2004self], the eigengap \$\lambda\_{K+1} - \lambda\_K\$ is used to evaluate clustering quality for \$K\$ clusters. Following a similar strategy, and assuming \$K\$ is known, the Laplacian eigengap at the \$K\$th eigenvalue can select \$m\$, with the scaling parameter fixed.
> >
> > Determining the optimal value of \$m\$ using the eigengap is not straightforward. As the affinity matrix dimension increases, the number of eigenvalues grows, reducing gaps between them. This makes direct comparisons unreliable. To address this, we use the ratio \$R = (\lambda\_{K+1} - \lambda\_K) / \lambda\_{K+1}\$. This metric is bounded between 0 and 1 and measures the relative difference between consecutive eigenvalues. It facilitates meaningful comparisons across different values of \$m\$. A value of \$R\$ close to 1 suggests high clustering quality, whereas lower values indicate weaker performance.
> >
> > Using this metric, we determine a near-optimal value for \$m\$ by maximizing the average \$R\$ across possible values of \$m\$. Additionally, the metric enhances robustness by running the algorithm with different random seeds and selecting the clustering result with the highest normalized eigengap.
> >
> > Zelnik-Manor, Lihi, and Pietro Perona. "Self-tuning Spectral Clustering." Advances in Neural Information Processing Systems 17 (2004).

---

> > ### Author Response · Authors · 2024-11-18
> > **Implementation of hyperparameter selection for the numver of Centroid in Python**
> >
> > We  used our new method `fit_select` which allows for the spectral bridges algorithm to be run multiple times, automatically selecting the best clustering result according to the normalized eigengap metric. Furthermore, it also provides the possibility to select the optimal number of cells $m$, by maximizing the mean normalized eigengap over a specified number of runs.

---

> ### Author Response · Authors · 2024-11-15
> **Hyperparameters: scaling**
>
> ####  Scaling parameter: Regularization
>
> We added in the paper:
>
> > This regularization is crucial: with a bounded affinity metric, exponentiation enhances the separation between low and high-density regions, controlled by a scaling parameter, as in traditional spectral clustering. Redefining the metric with a square root helps mitigate a converse issue, where machine error can cause numerical instability when solving the Laplacian eigenproblem, especially if values become too large. Since the range of affinity values can become too wide when the initial ratio between the largest and smallest non-zero unscaled bridge affinities is high, applying the square root reduces the maximum values in the affinity matrix while preserving the metric's interpretability and distance-like properties; importantly, this adjustment is not intended for outlier detection.

---

> ### Author Response · Authors · 2024-11-15
> **Remark about dimension reduction**
>
> #### Remark about dimension reduction
>
> The dimensionality in spectral clustering is not strictly a parameter of the method itself. We opted to illustrate some applications with dimensionality reduction because it is a common practice in clustering tasks. However, our proposed algorithm is designed to work directly with quantitative data without requiring dimensionality reduction.

---

> ### Author Response · Authors · 2024-11-15
> **Bridge affinity intuition**
>
> ### Criterion $B_{kl}$
>
> The spectral bridges criterion focuses on the data points between the two classes, specifically those projected onto the segment connecting the centers of the two clusters. These data points are significant because they provide insight into the boundary region between the classes, indicating where the separation is weakest.
>
> Each data point contributes proportionally to its square distance to the mediating hyperplane between the two classes. The distance to the mediating hyperplane is important because it reflects how closely a data point lies to the decision boundary, indicating its influence on the class separation.
>
> When classes are well separated, the overall contribution is small; however, when the classes are not well separated, the contribution becomes significant as it indicates areas of weak separation.
>
> The criterion is normalized to allow for comparisons between different bridges.
>
>
> The spectral bridges criterion thus implicitly considers the distance of each observation to the mediating hyperplane between the two cluster centers. The criterion evaluates the contribution of each data point based on how far it lies from the cluster pair boundary, which is effectively represented by the mediating hyperplane. By focusing on this distance, the criterion captures the influence of each point on the class separation. The form adopted by the criterion makes it computationally efficient while still reflecting the critical distances that affect it.
>
> An explanatory figure has been added (new Figure 2).

---

> ### Author Response · Authors · 2024-11-15
> **Figure 2**
>
> Thank you for pointing out the potential for confusion in old Figure 2.
>
> We changed the legend accordingly and it is more explicit:
>
> > Illustration of the Spectral Bridges algorithm with the Iris dataset (first principal plane). The bold red dots represent the centroids of the clusters, while the colored cells indicate the final partition of the data points. Vector quantization (Step 1 of Algorithm 1 ), Affinity computation (Step 2 of Algorithm 1 ), Spectral clustering and spreading (Step 3-4 of Algorithm 1 ).

---

> ### Author Response · Authors · 2024-11-15
> **Competing Methods**
>
> ###  Competing Methods
>
> We added
>
> - the GIT method (which was available in python) and
> - experiment after UMAP dimension reduction.
>
> We hope that these revisions will address your concerns and improve the manuscript. Thank you again for your constructive comments, which have greatly contributed to the quality of our work. We look forward to your feedback on the revised version.
>
> Best regards,
>
> Félix Laplante and Christophe Ambroise

---

### Comment · Action_Editor_tpEz · 2024-10-23
**Rebuttal period**

Dear Christophe Ambroise,

We have received two reports for your submission to Computo entitled “Spectral Bridges”.

A period of 6 weeks is allowed for discussion with the referees before they issue a final opinion. During this period, you can make any changes to your submission that you feel are necessary and that you are able to make. At the end of this period, a decision will be made, ranging from final acceptance to more substantial requests for modification.

We look forward to hearing from you, and thank you for choosing our journal,

---

> ### Comment · Action_Editor_tpEz · 2024-11-19
> **End of rebuttal period**
>
> Dear reviewers,
>
> I have received confirmation that the authors have responded to all your comments and produced a new version of the manuscript, available here :
> - https://github.com/cambroise/spectral-bridges-computo/ (github repo)
> - https://cambroise.github.io/spectral-bridges-computo/ (HTML/PDF manuscript)
>
> You can now make a recommendation based on the authors' responses and the updated version of the submission. On the basis of your recommendations, I will make a decision (acceptance, major or minor revisions, rejection).
>
> Thank you for your time in reviewing this manuscript.

---

### Note · Reviewer_pHXm · 2024-11-22

**Comment:**

Dear Julien,

I am fully satisfied with the authors' answers. They have taken my suggestions into account.
I think the article can be accepted in its current version.

Best,
Matt

**Audience:**

Yes

**Claims And Evidence:**

Yes

**Decision Recommendation:**

Accept

---

### Note · Reviewer_4dJq · 2024-12-12

**Comment:**

The authors have answered to all my remarks.

The article is now significantly improved and deserves publication.

**Audience:**

Yes

**Claims And Evidence:**

Yes

**Decision Recommendation:**

Accept

---

### Note · Action_Editor_tpEz · 2024-12-12

**Comment:**

Acceptance, based on reviewers' reports and modifications.

**Audience:**

Yes

**Claims And Evidence:**

Yes

**Decision Recommendation:**

Accept

---

### Decision · Action_Editor_tpEz · 2024-12-12

**Recommendation:** Accept as is

**Comment:**

The authors have addressed all the points raised by the reviewers.
I am pleased to inform you that your paper is accepted as is, and can now go in production.
Thank you for to the reviewers for their careful reading and for the authors for submitting to Computo.

**Audience:**

definitely appropriate to Computo's scope.

**Claims And Evidence:**

yes, based on the reviewers' report and to my personal reading

---

> ### Decision · Editors_In_Chief · 2024-12-12
>
> I approve the AE's decision.